Small molecule therapeutics for COVID-19: repurposing of inhaled furosemide

Wang Zhiyu 1 2
Wang Yanfei 1
Vilekar Prachi 1
Yang Seung-Pil 1
Gupta Mayuri 1
Oh Myong In 1
http://orcid.org/0000-0002-6902-4620 Meek Autumn 1
Doyle Lisa 1
http://orcid.org/0000-0003-4311-3278 Villar Laura 1
Brennecke Anja 1
Liyanage Imindu 1 3
Reed Mark 1 4
Barden Christopher 1
Weaver Donald F. 1 2 3 dweaver@uhnres.utoronto.ca
1 Krembil Research Institute, University Health Network , Toronto, ON , Canada
2 Faculty of Pharmacy, University of Toronto , Toronto, ON , Canada
3 Faculty of Medicine, University of Toronto , Toronto, ON , Canada
4 Department of Pharmacology and Toxicology, University of Toronto , Toronto, ON , Canada
Gomez Shawn
Electronic publication date: 2020 Jul 7
Publication date: 2020
Volume: 8
Electronic Location ID: e9533
Received 2020 Apr 13; Accepted 2020 Jun 23
Copyright: © 2020 Wang et al.
Copyright year: 2020
Copyright holder: Wang et al.
License: This is an open access article distributed under the terms of the Creative Commons Attribution License, which permits unrestricted use, distribution, reproduction and adaptation in any medium and for any purpose provided that it is properly attributed. For attribution, the original author(s), title, publication source (PeerJ) and either DOI or URL of the article must be cited.
License URL: https://creativecommons.org/licenses/by/4.0/

Keywords: Furosemide, COVID-19, Cytokine storm, Hypercytokinemia, Coronavirus, Anti-inflammatory

Funding: Canada Research Chair Program (Donald F. Weaver, Canadian Research Chair Tier 1) This work was supported by the Canada Research Chair Program (Donald F. Weaver, Canadian Research Chair Tier 1). The funders had no role in study design, data collection and analysis, decision to publish, or preparation of the manuscript.

==============================
The novel coronavirus SARS-CoV-2 has become a global health concern. The morbidity and mortality of the potentially lethal infection caused by this virus arise from the initial viral infection and the subsequent host inflammatory response. The latter may lead to excessive release of pro-inflammatory cytokines, IL-6 and IL-8, as well as TNF-α ultimately culminating in hypercytokinemia (“cytokine storm”). To address this immuno-inflammatory pathogenesis, multiple clinical trials have been proposed to evaluate anti-inflammatory biologic therapies targeting specific cytokines. However, despite the obvious clinical utility of such biologics, their specific applicability to COVID-19 has multiple drawbacks, including they target only one of the multiple cytokines involved in COVID-19’s immunopathy. Therefore, we set out to identify a small molecule with broad-spectrum anti-inflammatory mechanism of action targeting multiple cytokines of innate immunity. In this study, a library of small molecules endogenous to the human body was assembled, subjected to in silico molecular docking simulations and a focused in vitro screen to identify anti-pro-inflammatory activity via interleukin inhibition. This has enabled us to identify the loop diuretic furosemide as a candidate molecule. To pre-clinically evaluate furosemide as a putative COVID-19 therapeutic, we studied its anti-inflammatory activity on RAW264.7, THP-1 and SIM-A9 cell lines stimulated by lipopolysaccharide (LPS). Upon treatment with furosemide, LPS-induced production of pro-inflammatory cytokines was reduced, indicating that furosemide suppresses the M1 polarization, including IL-6 and TNF-α release. In addition, we found that furosemide promotes the production of anti-inflammatory cytokine products (IL-1RA, arginase), indicating M2 polarization. Accordingly, we conclude that furosemide is a reasonably potent inhibitor of IL-6 and TNF-α that is also safe, inexpensive and well-studied. Our pre-clinical data suggest that it may be a candidate for repurposing as an inhaled therapy against COVID-19.

Introduction

COVID-19, a potentially lethal infection caused by the SARS-CoV-2 virus, has emerged as a public health crisis of global concern. Currently, there are no effective curative treatments for COVID-19, affording little patient recourse beyond supportive care (Cortegiani et al., 2020). The morbidity and mortality of COVID-19 arise from two competing pathological processes, the initial viral infection and the subsequent host inflammatory response, the latter of which may lead to excessive release of pro-inflammatory cytokines (interleukins (e.g., IL-6, IL-8) or non-interleukins (e.g., TNF-α)) and may culminate in hypercytokinemia (“cytokine storm”), a self-targeting inflammatory response syndrome (Conti et al., 2020; Mehta et al., 2020; Qin et al., 2020; Huang et al., 2020). Reflecting these dichotomous disease processes, current therapeutic development strategies may be categorized into two broad groups: anti-viral and anti-inflammatory. Arguably, truly effective treatments may require both agents, since an antiviral will fail to suppress uncontrolled pro-inflammatory cytokine release once the process has been triggered.

To address the COVID-19 immuno-inflammatory pathogenesis, multiple clinical trials have been proposed to evaluate anti-inflammatory biologic therapies targeting specific cytokines. Agents under consideration include tocilizumab (Peking University First Hospital, 2020; Tongji Hospital et al., 2020; National Cancer Institute Naples, 2020) and sarilumab (Regeneron Pharmaceuticals & Sanofi, 2020), both monoclonal antibodies that target the IL-6 pathway (Sebba, 2008; Boyce et al., 2018), as well as adalimumab, which binds with specificity to TNF-α (Furst et al., 2003). Data from a Chinese study in which tocilizumab was given to 21 patients with severe COVID-19 reported that the patients “improved remarkably” with 13 patients being discharged within 14 days or less following treatment, six patients within 14–21 days and only two patients stayed in the hospital for more than 21 days (Xu et al., 2020). However, despite the obvious clinical utility of such biologics for many disorders, their specific applicability to COVID-19 is hampered by various issues: they target only one of the multiple cytokines implicated in COVID-19’s immunopathy; if administered systemically, they can predispose patients to secondary infections or other toxicities, such as hepatoxicity; and they may be expensive to mass produce and distribute. Therefore, they are of limited utility in the context of a global pandemic.

Accordingly, we set out to identify a small molecule with the following properties: broad-spectrum anti-inflammatory mechanism of action targeting cytokines of innate immunity; low toxicity and excellent safety profile; chemically stable; easily stored and administered; able to be rapidly adopted in clinical settings worldwide; and, widespread availability with inexpensive and efficient means of production. As described herein, a systemic series of in silico and in vitro studies have enabled us to identify furosemide as a candidate molecule.

Materials and Methods

In silico studies

Molecular docking simulations

Molecular docking simulations on the endogenous molecule (1,136, created in-house through literature search) and known drug datasets (1,768, taken from drugbank.ca) against Il-6 and TNF-α were carried out in two steps. The first step involved placement of the endogenous or ligand molecule in an identified binding pocket (docking) of IL-6 and TNF-α. In the second step, the calculation of the binding energies of the docked molecules (scoring) was completed.

Preparation of TNF-α and IL-6 3D protein structures

The TNF-α structure (PDB code: 2AZ5) and IL-6 structure (PDB code: 1ALU) were selected for molecular docking simulations in this work. We have employed these X-ray crystallographic structures because of three underlying motivations: (1) These X-ray crystallographic structures are solved for human IL-6 and TNF-α proteins. (2) Both of these X-ray structures have been solved at high resolution that is, 1 ALU for IL-6 at 1.90 Å and 2AZ5 for TNF-α at 2.10 Å, which minimizes the error while solving the structure from electron density maps. (3) The ligand molecules did not involve any metal atoms, which can otherwise bias the active site binding residues.

The downloaded PDB structures were loaded into Molecular Operating Environment (MOE 2019.0101) software (Molecular Operating Environment (MOE), 2019). The loaded crystal structure was prepared employing the “QuickPrep Panel” in MOE, which contains the “structure preparation” feature. The “Protonate3D” function was used to optimize ionization states of the added hydrogen atoms. Water molecules, which were present 4.5 Å away from ligand or receptor, were deleted. The MFF94x force field, with RMS gradient of 0.05 kcal/Å, was used for energy minimization (Panigrahi & Desiraju, 2007). A Ramachandran plot was used to confirm the geometric and stereochemical qualities of the TNF-α and IL-6 protein structures.

Endogenous molecules and known drugs dataset curation

The datasets of endogenous molecules (1,136) and drugs (1,768) were prepared using MOE software. Two functions ‘wash at dominant pH of 7.4′ and “energy minimize” were used to obtain an energy minimized molecular form of molecules present at physiological pH of 7.4.

Binding site selection and docking simulations

The active site of IL-6 and TNF-α was identified using the “MOE-site Finder” module. The “Triangle Matcher” placement method and “London dG” scoring functions were used for docking simulations. The poses from ligand conformations were generated in the Placement phase. After placement of poses, the refinement was done with the “GBVI/WSA dG” scoring method using “induced fit” option. Induced fit refinement allows both the ligand molecule and protein active site to move freely to facilitate residue alignment during docking simulations. Thirty docked poses of the placement retention simulation and 10 poses of the refinement step were retained. The docking pose corresponding to the highest score for an endogenous or a drug molecule was considered for comparison of binding affinity within datasets. Further details of descriptor calculation and docking simulations are provided in our recent publications (Gupta et al., 2019, 2020) and in supporting information.

Docking method validation

Validation of the docking protocol was carried out by re-docking the co-crystallized ligand in the active sites of the IL-6 and TNF-α protein receptors. The following parameters were confirmed for validating the accurate docking protocol:Molecular superposition: The lowest energy docked conformation of a native ligand for IL-6 and TNF-α is overlaid relative to the co-crystallized ligand present in the X-ray structure of the protein; the root-mean-square deviation (RMSD) has been calculated and is taken as an evaluation criterion for the validation of docking simulations. An RMSD value of <2.0 Å between docked and crystallographic ligand conformation confirms successful docking (Plewczynski et al., 2011; Ramírez & Caballero, 2018).

Ligand interactions with the active site: The docking protocol is validated if the docked ligand interacts with the active site of the protein similar to the corresponding co-crystallized ligand conformer and exhibits the same interactions with active site amino acid residues (Plewczynski et al., 2011; Ramírez & Caballero, 2018).

In vitro studies

Inflammation activation on RAW264.7 macrophage

RAW264.7 cells were purchased from ATCC and maintained in Dulbecco’s Modified Eagle’s Medium (DMEM) containing foetal bovine serum (FBS) at a final concentration of 10%. RAW264.7 cells were seeded in 12-well plates at 0.25 × 106 cells/well seeding density, one day before experiments. To activate the cells, cell culture medium was changed to a lipopolysaccharide (LPS) and Interferon γ (IFNγ) containing medium with dimethyl sulfoxide (DMSO) or furosemide at required concentrations, followed by 24 h incubation at 37 °C; conditioned media and lysates were harvested for analysis.

Inflammation activation on THP-1 monocytic cells

THP-1 cells (ATCC) were maintained in RPMI 1,640 medium supplemented with 2-mercaptoethanol at a final concentration of 0.05 mM and FBS at a final concentration of 10%. THP-1 cells were seeded in each well of a 12-well tissue culture plate at a density 0.5 × 106 cells/mL, one day before experiments. THP-1 monocytes were differentiated by 150 nM PMA (phorbol 12-myristate 13-acetate) for 24 h. Cells were then treated with LPS+IFNγ with DMSO or furosemide at the required concentration, followed by 48 h incubation; conditioned media and lysates were collected for analysis.

Nitric oxide production by Griess assay

Nitric oxide production from the conditioned media of RAW264.7 and SIM-A9 cell cultures was examined by a Griess assay. Conditioned medium and sulfanilamide were mixed in a microwell plate to form a transient diazonium salt. Then, N-naphtyl-ethylenediamine was added to all wells to form a stable azo compound by incubating for 5–10 min in the dark at room temperature. The absorbance was measured between 520 nm and 550 nm. The concentration of NO production was quantified by being plotted against a standard curve.

Enzyme-linked immunosorbent assay

Cytokines were quantified using ELISA kits following the manufacturer’s instructions. Briefly, the high-binding plates were coated at 100 μL/well with diluted capture antibodies (1:250) at 4 °C overnight. The coated plates were then blocked with the diluent for 1 h before assay. Each sample was diluted accordingly and added to the plates for a 2 h incubation period at room temperature. Plates were then washed with 250 μL/well PBS with 0.05% Tween-20 and incubated with detection antibodies (1:250 in assay diluent) for 1 h at room temperature. After another washing step, 1:250 diluted avidin-HRP (horseradish peroxidase) was added and incubated for 30 min. Next, 100 μL TMB-substrate (3,3′,5,5′-tetramethylbenzidine) was added and the plate was incubated in the dark until the signal was sufficiently developed. The final concentration of TMB substrate solution was 200 µg/mL. The reaction was stopped with 50 μL of 2 N sulfuric acid. Absorbance was measured at 450 nm with a correction of 570 nm using a plate reader.

Western blotting

Cells were washed twice with ice-cold PBS and harvested in RIPA buffer supplemented with a protease inhibitor cocktail. The whole-cell extracts were then centrifuged at 22,000×g for 20 min at 4 °C to remove cell debris. Protein concentrations were quantified using a Micro BCA protein assay kit. The absorbance was measured at 595 nm using a microplate reader. Equal amounts of cellular protein were separated by sodium dodecyl sulfate-polyacrylamide gel electrophoresis (SDS-PAGE) and transferred onto polyvinylidene difluoride (PVDF) membranes at 100 V for 90 min. The SDS-PAGE was performed with 10% polyacrylamide gel for iNOS and 12.5% for pro-IL-1β. The membranes were blocked for 1 h in Tris-buffered saline (TBS), pH 7.4, with 0.1% Tween-20 (TBS-T) containing 10% skim milk. The membrane blot was then incubated overnight at 4 °C with primary antibodies against iNOS (1:1,000), IL-1β (1:1,000), actin (1:5,000) and GAPDH (1:5,000) in TBS-T containing 5% skim milk. The membrane was washed with TBS-T 3 × 10 mins and incubated with goat anti-rabbit IgG-horseradish peroxidase (1:5,000) for 1 hour. After the washing step, the immunoblotting was visualized by chemiluminescence HRP-substrate.

Flow cytometry

Cells were harvested and re-suspended with staining buffer. Cells were stained with antibody (1:100) by incubating at 4 °C for 30 min in the dark. Stained cells were centrifuged, and the supernatant was discarded. The cell pellets were then re-suspended in cell flow buffer, transferred to FACS tubes and analyzed by flow cytometry within 48 h. To all cells in the experiment, Fc blocker was added.

Inflammation activation on SIM-A9 cells

SIM-A9 (ATCC) cells were maintained in Dulbecco’s modified eagle medium: nutrient mixture F-12 (DMEM-F12) with 10% foetal bovine serum, 5% horse-serum and antibiotic–antimycotic. SIM-A9 cells were seeded 24 h before the experiment. Culturing medium was replaced with DMEM-F12 medium containing 5% FBS + 2.5% horse serum with required LPS concentration (final volume was one mL/well). The conditioned media and lysates were harvested for cytokine and cell marker examination.

Statistical analysis

Data are presented as mean ± SD or ±SEM. Statistical analysis was performed with GraphPad Prism software version 6.01c, applying a two-tailed unpaired t-test. A p-value of >0.05 was considered significant.

Results

Identifying an initial hit

A screening strategy was devised for identifying an initial compound with potential broad-spectrum anti-inflammatory activity targeting relevant cytokines of the pulmonary innate immune system. Since immunologically-mediated inflammatory responses in the human body are subject to tight homeostatic regulation, there exist multiple endogenous compounds capable of either up- or down-regulation of innate immune processes. Accordingly, we sought to identify an endogenous compound as an initial molecular platform around which to devise a therapeutic. A library of 1,136 small molecules endogenous to the human body was assembled, subjected to in silico molecular docking simulations and a focused in vitro screen to identify anti-pro-inflammatory activity via interleukin inhibition.

Multiple compounds within the tryptophan metabolic pathway were identified, both indoleamine and anthranilate metabolites. In particular, 3-hydroxyanthranilic acid (3-HAA) was found to demonstrate significant anti-inflammatory potential (in accord with previous studies by others, such as Lee et al. (2013), who have shown significant activity of 3-HAA in inhibiting IL-6 and TNF-α). Regrettably, 3-HAA is a small polar molecule with poor drug-like properties and is not an approved therapeutic for human use.

Converting hit to drug-like anthranilate compound

The task of converting 3-HAA to a drug or drug-like compound can be addressed via two approaches: (1) synthesis of new chemical entities with drug-like properties based on the 3-HAA scaffold, or (2) identification of known drugs with structural properties similar to the 3-HAA scaffold with the goal of repurposing. Because of the urgency imposed by the unfolding pandemic, approach 2 was selected to provide a short-term solution.

Accordingly, 1,768 known drugs were computationally evaluated for anthranilate structural components. This screen identified mefenamic acid (N-(2,3-xylyl)-anthranilate) and furosemide (4-chloro-5-sulfamoyl-N-furfuryl-anthranilate) as the two strongest leads. Mefenamic acid is known to provide significant protection against elevated levels of TNF-α and IL-1β in radiation-induced genotoxicity of human lymphocytes (Armagan et al., 2012; Hosseinimehr et al., 2015); furosemide is known to significantly reduce production of IL-6, IL-8 and TNF-α in bronchial inflammation of asthma (Prandota, 2002; Yuengsrigul, Chin & Nussbaum, 1999). This screen also identified other loop diuretics structurally related to 3-HAA and furosemide, including bumetanide, piretanide and azosemide (Fig. 1).

Figure 1 Screening approach with chemical structures of important compounds.

With the goal to find a small therapeutic with anti-inflammatory properties and endogenous to the human body, a library of 1,136 small molecules was screened by docking simulations and further investigated by in vitro experiments. This intensive screen yielded 3-hydroxyanthranilic acid (3-HAA) as initial hit. Since 3-HAA is not approved for medical applications, 1,768 known drugs were screened computationally for a 3-HAA substructure. Besides mefenamic acid, which is not displayed in this figure, the diuretic furosemide was the strongest lead directly followed by bumetanide, azosemide and piretanide.

Arising from the observation that furosemide can reduce bronchial inflammation, can be administered by inhalation (vide infra, Discussion) (Prandota, 2002; Inokuchi et al., 2014; Waskiw-Ford et al., 2018) and is a widely available drug, furosemide was selected to be explored for repurposing in the treatment of COVID-19. Furosemide has been used worldwide for decades as a loop diuretic working via the renal Na+/K+-ATP shuttle pathway.

In vitro assessment of lead anthranilate compound

To pre-clinically evaluate furosemide as a putative COVID-19 therapeutic, a series of in vitro efficacy assessments was performed to investigate its anti-inflammatory properties. First, we investigated if furosemide could reduce the release of pro-inflammatory cytokines induced by LPS in macrophage cell line. RAW264.7 macrophages were stimulated with LPS in the presence or absence of furosemide and the levels of TNF-α and NO were measured from the conditioned media using ELISA and Griess assay, respectively. The results in Figs. 2A and 2B show that LPS induces the production of NO and TNF-α, indicating macrophage polarization to an M1 pro-inflammatory phenotype.

Figure 2 Furosemide decreases the production of NO and TNF-α.

Production of (A) NO and (B) TNF-α from RAW264.7 cells upon LPS induction was determined by Griess assay and ELISA from the conditioned medium. Error bars show SEM, n = 3. *, p < 0.05. (C) iNOS expression level from RAW264.7 cells was assessed by Western blot analysis. Cells were treated with LPS+IFNγ with DMSO or 25 μM furosemide. After 24 h of incubation, cell lysate was harvested for Western blot analysis. GAPDH, Glyceraldehyde 3-phosphate dehydrogenase.

When cells were treated with LPS in the presence of 25 μM of furosemide, the production of NO and TNF-α significantly decreased. To further investigate the reduction of NO production by furosemide, we determined the expression level of inducible nitric oxide synthase (iNOS) which produces NO in response to inflammatory stimulations. Therefore, we stimulated RAW264.7 macrophages with LPS and IFNγ. The expression of iNOS was significantly induced by stimulation with LPS and IFNγ, as shown by the Western blot results in Fig. 2C. We found that furosemide was able to suppress the expression of iNOS during LPS and IFNγ induced stimulation. Densitometry analysis showed that normalized iNOS/GAPDH ratio was reduced from 1 to 0.88 by furosemide.

LPS is recognized by the cell surface pattern-recognition receptors such as the toll-like receptor 4 protein (TLR4) and triggers downstream signaling pathways. LPS induces expression of the pro-inflammatory cytokine IFNγ. IFNγ is known to increase TLR4 expression which may then promote the response to LPS stimulation. We investigated the effect of furosemide on LPS-induced TLR4 expression in RAW264.7 macrophage cells using flow cytometry. As shown in Fig. 3, LPS increased the TLR4+ cell population significantly, whereas IL-4, an anti-inflammatory cytokine, barely induced TLR expression from the cells. Interestingly, furosemide completely blocked LPS-induced TLR4 expression, suggesting the possible involvement of furosemide in the LPS- or IFNγ-induced inflammation.

Figure 3 Furosemide significantly decreases TLR4+ cell population.

RAW264.7 macrophage cells were stimulated with LPS and flow cytometry was used to determine TLR4+ cells population. (A) % of Vis and (B) cell numbers for TLR4+. F, furosemide, FSC-A, front scatter.

As the next stage of macrophage activation, we studied the activity of furosemide on the expression of IL-1β which plays a key role in modulating inflammatory response, as a downstream pro-inflammatory marker of the TLR4 signaling pathway by using differentiated THP-1 monocytes. We analyzed the expression of IL-1β precursor protein (pro-IL-1β) from THP-1 macrophage cells by using Western blot analysis. Furosemide, as shown in Fig. 4, significantly decreases the expression of pro-IL-1β in differentiated THP-1 cells, demonstrating yet another inflammatory cytokine targeted by furosemide. To further explore the effect of furosemide on different macrophages, we next tested it on SIM-A9 cells. We stimulated SIM-A9 macrophages with LPS and analyzed the release of pro-inflammatory markers such as NO, IL-6 and TNF-α. The results in Fig. 4 show that LPS induced the production of NO, IL-6 and TNF-α from SIM-A9 cells. Similar to RAW264.7 cell results, furosemide significantly reduced the production of all these pro-inflammatory markers from SIM-A9 cells as well.

Figure 4 Furosemide significantly inhibits pro-inflammatory responses.

Furosemide activity was initially tested on differentiated THP-1 macrophages. THP-1 monocytic cells were initially differentiated to THP-1 macrophages by PMA for 24 h. Cells were then stimulated by LPS/IFNγ in the presence of furosemide or DMSO control. Followed by 48 h incubation, the conditioned media and cell lysates were harvested for analysis. (A) Western blot showing the expression of pro-IL-1β upon treatment with DMSO and furosemide, respectively. Actin was used as a loading control. Productions of (B) NO, (C) IL-6 and (D) TNF-α from SIM-A9 cells were measured by either Griess assay or ELISA. Error bars show SD, n = 6. **, p < 0.01; ***, p < 0.001.

We have tested three different macrophage cell lines for the effects of furosemide on pro-inflammatory markers. Furosemide consistently reduced pro-inflammatory markers such as NO production, secretion of IL-6, TNF-α and expression of pro-IL-1β from different macrophage cell lines, implying that furosemide has broad inhibitory activity against pro-inflammatory cytokines.

Then, we evaluated if furosemide exhibits any effects on anti-inflammatory cytokines. We measured the levels of anti-inflammatory cytokines from the conditioned medium of differentiated THP-1 cells after 48 h of stimulation with LPS and IFNγ by Multiplex assay using flow cytometry. Furosemide induces the expression of IL-4, interleukin-1 receptor antagonist (IL-1RA) and arginase, which are anti-inflammatory markers, suggesting the polarization of THP-1 macrophage to an M2 phenotype (Fig. 5).

Figure 5 Furosemide induces the expression of anti-inflammatory phenotype markers on THP-1 macrophages.

The anti-inflammatory activity of furosemide was tested by THP-1 macrophages. THP-1 monocytic cells were initially differentiated to THP-1 macrophage by PMA for 24 h. Cells were then stimulated by LPS/IFNγ in the presence of furosemide or DMSO control. Followed by 48 h incubation, the conditioned media were harvested for (A) IL-4, (B) IL-1RA and (C) Arginase analysis. Error bars show SD, n = 2. **, p < 0.01.

Together with the experiments discussed above, these results show that furosemide inhibits the expression of M1 pro-inflammatory markers and promotes the expression of M2 anti-inflammatory markers. Thus, furosemide is a broad-spectrum anti-inflammatory drug candidate targeting multiple cytokines.

In silico mechanism of action simulations

The molecular mechanism of action whereby 3-HAA and furosemide inhibit IL-6 and TNF-α activity is unknown at this time and lies outside the scope of the present study. Interestingly, in an in silico screen of possible binding sites for 3-HAA and related drug anthranilates, we noted favorable interactions with both the IL-6 and TNF-α proteins. Whilst it is improbable that such interactions are completely responsible for the observed anti-inflammatory effects, this notable observation is worthy of future experimental evaluation.

The docking protocol needs to be validated before any conclusive in silico results can be obtained. To validate the docking methodology, co-crystalized ligands for TNF-α and IL-6 have been re-docked into the active sites of the corresponding proteins. Table 1 shows the re-docking parameters that have been evaluated for confirmation of accuracy of the employed docking protocol. IL-6 (PDB code: 1ALU) has been re-docked with tartaric acid.

Table 1 Re-docking of the co-crystallized ligands tartaric acid and 307* in the active site of IL-6 and TNF-α.

The table includes RMSD data, interacting amino acid residues and binding energy values.

Protein	PDB ID	Co-crystallized ligand	Interacting amino acid residues in the active site	Binding pocket residues within 4 Å of radius	RMSD (Å)	Binding energy (kcal/mol)	
IL-6	1ALU	Tartaric acid	Gln175
Arg179
Arg182	Arg30
Leu178	0.6114	−4.7397	
TNF-α	2AZ5	Bound inhibitor-307* in TNF-a dimer complex form	Tyr119	Leu57
Tyr59
Ser60
Leu120
Gly121
Gly122
Tyr151	0.1677	−7.5526	
Note:

* TNF-α dimer inhibitor 307 IUPAC name: 6,7-DIMETHYL-3-[(METHYL{2-[METHYL({1-[3-(TRIFLUOROMETHYL)PHENYL]-1H-INDOL-3-YL}METHYL)AMINO]ETHYL}AMINO) METHYL]-4H-CHROMEN-4-ONE.

After re-docking simulations, the docked conformer of tartaric acid is overlaid with the co-crystallized ligand in the PDB structure. Figure 6A represents the overlay of the docked pose and the co-crystallized ligand conformer. The RMSD between the docked conformer and co-crystallized ligand is 0.6114 Å, which is <2.5 Å as recommended for successful docking method confirmation. Furthermore, all the amino acid residues which are interacting with the native IL-6 binding pocket (Arg179, Arg182 and Gln175) are shown to be interacting with the docked conformer of the tartaric acid in the re-docking simulation (as shown in Fig. 6B).

Figure 6 Docking simulations of co-crystallized ligand in IL-6.

(A) The overlay of docked conformer of tartaric acid against co-crystallized ligand conformation in IL-6 active site for validation of docking simulations. (B) Ligand interaction diagram of co-crystallized ligand tartaric acid of IL-6 (PDB code: 1ALU) in re-docking simulation.

This further confirms that the re-docking method is able to accurately predict the binding mode and the active site of IL-6. The TNF-α dimer inhibitor has also been re-docked with TNF-α for validation of the docking method. The overlay of the docked ligand with the co-crystallized ligand conformation is presented in Fig. 7A. From Fig. 7A it can be seen that the TNF-α dimer inhibitor 307 is superposed with the co-crystallized ligand conformer with an RMSD value of 0.1677 Å, which validates the docking simulation methodology for TNF-α. The ligand interaction diagram of the docked conformer of TNF-α, is shown in Fig. 7B. The amino acid residues present within 4 Å of the active site are labeled i.e. Leu57, Tyr59, Ser60, Leu120, Gly121, Gly122, Tyr151. These residues are present in the active site of the TNF-α dimer in the vicinity of co-crystallized ligand. The re-docking simulations of co-crystallized ligands for IL-6 and TNF-α confirm the accuracy of the docking methodology employed in this study.

Figure 7 Docking simulations of co-crystallized ligand in TNF-α.

(A) The overlay of docked conformer of TNF-α dimer inhibitor 307 against co-crystallized ligand conformation in TNF-α active site for validation of docking simulations. (B) Ligand interaction diagram of co-crystallized ligand of TNF-α dimer in a re-docking simulation.

The computational simulations of 3-HAA and furosemide with TNF-α and IL-6 are briefly outlined below. The binding of 3-HAA in IL-6 and TNF-α active sites is presented in Fig. 8. The docking score (S) of minimum energy pose of 3-HAA in IL-6 is −4.620 and in TNF-α is −4.487. In Figs. 8A and 8C, the ligand 3-HAA is shown in yellow and amino acid residues of binding pocket have been labeled. Figures 8B and 8D represent the ligand interaction diagrams of 3-HAA with active site residues of TNF-α and IL-6. 3-HAA interacts with the Arg179 and Arg182 in IL-6 and docks in the TNF-α dimer site with residues Tyr119, Tyr59, Ser60, Leu120, Gly121 and Tyr151. This confirms that 3-HAA interacts with IL-6 and TNF-α in the same binding site as observed in re-docking simulations of co-crystallized ligands for the corresponding proteins.

Figure 8 Interaction of 3-HAA with active site of TNF-α and IL-6.

(A) Binding of 3-HAA in the active site of tumor necrosis factor α (TNF-α); (B) Ligand interaction diagram of 3-HAA in binding site of TNF-α. (C) Binding of 3-HAA in the active site of interleukin-6 (IL-6); (D) Ligand interaction diagram of 3-HAA in binding site of IL-6.

Figure 9 presents docking simulations of furosemide with TNF-α and IL-6, respectively. In Figs. 9A and 9C, furosemide is shown in yellow and the amino acid residues of the binding site are labeled. The TNF-α protein complex (PDB: 2AZ5) contains four identical chains having 148 amino acids each and two bound ligands. The A and B chains were retained (coloured orange and purple) and the co-crystalized ligand was removed while preparing the protein structure for docking.

Figure 9 Interaction of furosemide with active site of TNF-α and IL-6.

(A) Binding of furosemide in the active site of tumor necrosis factor α (TNF-α); (B) Ligand interaction diagram of furosemide in binding site of TNF-α. (C) Binding of furosemide in the active site of interleukin-6 (IL-6); (D) Ligand interaction diagram of furosemide with binding site of IL-6.

Figure 9A shows the minimum energy docking pose of furosemide in the TNF-α active site. The amino acid residues Leu B57, Tyr B59, Gly B121, Tyr A151, Tyr A119 and Leu A120 are involved in forming hydrogen bonding and π-π interactions (arene–arene, arene–H, arene–cation) with furosemide in minimum energy docked poses. The docking score of the minimum energy pose is −6.0854. The similar mode of binding in the active site of TNF-α as of co-crystallized ligand (as shown in Fig. 7B) and favorable binding energy indicates that furosemide fits well into the active site of TNF-α and inhibits its activity.

Figures 9C and 9D show the minimum energy docking pose of furosemide in the protein structure of human IL-6 (PDB: 1ALU), having 186 amino acid residues and a co-crystallized ligand (tartaric acid). The co-crystallized ligand had been removed during the structure preparation step of the IL-6 protein. Furosemide interacts with Arg 182, Gln 175, Leu 33 and Arg 179 in most of the minimum energy poses, which agrees well with the IL-6 co-crystallized ligand tartaric acid binding mode. The docking score (S) of the minimum energy pose of furosemide with IL-6 is −5.1343, which computationally supports the ability of furosemide to inhibit IL-6 activity.

Discussion

The pathogenic mechanisms of COVID-19 morbidity and mortality are diverse, though immuno-inflammatory contributions are likely a central player. It is now appreciated that COVID-19 afflicted individuals with major respiratory symptoms have pathologically elevated levels of pro-inflammatory cytokines including IL-6, IL-8 and TNF-α (Conti et al., 2020; Mehta et al., 2020; Qin et al., 2020; Huang et al., 2020). A logical therapeutic approach to the management of COVID-19 thus includes a need to modulate immunotoxicity.

Tryptophan and its metabolites, particularly via the indoleamine-2,3-dioxygenase initiated pathway, have a previously described role as endogenous modulators of innate immunity. Of these various metabolites, 3-HAA has been identified to exhibit a significant anti-inflammatory ability to suppress inflammation mediated via multiple pro-inflammatory interleukin cytokines, including IL-1β, IL-6, IL-8 and TNF-α (Lee et al., 2013). This motivated our search for an anthranilate-based 3-HAA-like agent from a library of known drugs; furosemide emerged as a possible candidate from this search.

In this study, as part of its pre-clinical evaluation, we studied furosemide’s anti-inflammatory activity on multiple macrophage cell lines involved in innate immunity. As pattern recognition receptors, TLRs contribute to the recognition of the molecules that are commonly shared by pathogens, such as LPS and viral nucleotides (Alexopoulou et al., 2001). The activation of TLRs triggers downstream signaling through the MyD88-dependent pathway and eventually induces the activation of nuclear factor-кB (NF-кB) (Oeckinghaus, Hayden & Ghosh, 2011). NF-кB has been shown to play an important role in coronavirus infections. For instance, NF-кB activation was identified in the lungs of SARS-CoV infected mice and triggered the production of pro-inflammatory cytokines, such as TNF-α (DeDiego et al., 2014). However, the mechanism of SARS-CoV pathogenesis is debated. An in vitro study suggested the nucleocapsid protein of SARS-CoV was crucial in the pathogenesis, although this correlation is cell-specific (Liao et al., 2005). On the other hand, SARS-CoV lacking the envelope protein (E protein) attenuated NF-кB activation and associated pro-inflammatory responses (DeDiego et al., 2014). The pandemic outbreak of COVID-19 is caused by the infection of SARS-CoV-2 that shares 79.5% identity to SARS-CoV (Guo et al., 2020). To investigate the potential use of furosemide as a therapy for COVID-19, we developed in vitro assays using LPS as exogenous stimulation to induce inflammatory responses. LPS-induced TLRs activation is well studied: it interacts with TLRs and triggers NF-кB activation followed by pro-inflammatory responses (Lu, Yeh & Ohashi, 2008). In our in vitro assays, we aimed to ascertain if LPS induces a similar “cytokine storm” as SARS-CoV-2 infection and if furosemide inhibits the production of pro-inflammatory cytokines or promotes the secretion of anti-inflammatory cytokine products.

Broadly conceptualized stimulated macrophages can be polarized to either an M1 pro-inflammatory phenotype or an M2 anti-inflammatory phenotype. We investigated if furosemide inhibits the production of pro-inflammatory cytokines (M1) or promotes the secretion of anti-inflammatory cytokines (M2) using various macrophage cell lines including RAW264.7, THP-1 and SIM-A9. Upon stimulation, these cell lines initiated an immune response by producing cellular stress signals and secreting pro-inflammatory cytokines. These pro-inflammatory markers were reduced upon treatment with furosemide, indicating that furosemide suppresses the M1 polarization, including NO, IL-6 and TNF-α. More importantly, our study results demonstrated furosemide promotes the production of anti-inflammatory cytokine products (IL-1RA, arginase), indicating M2 polarization. All these results strongly suggest the potential application of furosemide as an immunomodulating agent for such disease conditions in which the inflammatory burden of patients increases suddenly. It also suggests that furosemide can be used for treating COVID-19 in which the sudden increase of pro-inflammatory cytokines is part of the disease pathogenesis.

Furosemide is a small molecule with a molecular weight of 330.75 g/mol and relatively low lipophilicity (logP = 2.03) (Hardman, Goodman & Gilman, 2001). Although the drug has low water solubility at pH 7, furosemide can be formulated in weakly basic buffer solution (pH 8) to achieve 10 mg/mL solutions suitable for intravenous administration. Due to the presence of a primary sulfonamide and carboxylic acid group, furosemide is highly bound to albumin with a human plasma protein binding value of 98.6 ± 0.4%. The drug has a very low volume of distribution (VD = 0.13 ± 0.06 L/kg) and a relatively short half-life of 1.3 ± 0.8 h (Hardman, Goodman & Gilman, 2001). These pharmacokinetic parameters along with high plasma protein binding equates to low tissue distribution with furosemide being retained with the blood. Alveolar macrophages are front line innate immune cells, playing a crucial role in the maintenance of lung homeostasis and lung tissue defense through various immune responses. Tissue-resident alveolar macrophages are derived from the foetal liver and populate the alveoli shortly after birth. These macrophages are self-renewing and persist over the life span. However, exposure to environmental challenges and injury induces recruitment of monocyte-derived alveolar macrophages from circulating monocytes. The intrinsic molecular properties of furosemide are ideal for targeting macrophages recruited from the blood stream prior to their tissue distribution.

Furosemide exhibits a large therapeutic window and is listed on the WHO’s List of “Essential Medicines”; it is readily available worldwide, is easily manufactured, and has a long record of safety and efficacy when given orally or intravenously. More importantly, furosemide may also be administered safely by inhalation. More than 20 years ago, the concept of inhaled furosemide was explored as an approach to reduce dyspnea, primarily based on the rationale that edematous airway mast cells would be reduced in size following diuresis (Prandota, 2002). However, further investigations established that the mechanism of action was not related to local diuretic effects or engagement of the Na+/K+-ATP shuttle. More recent studies have reported reduction in pulmonary IL-6, IL-8 and TNF-α levels upon administering inhaled furosemide to patients with conditions including tachypnea (Armed Forces Hospital Pakistan, 2016; University of Cologne, 2012), bronchopulmonary dysplasia (University of North Carolina et al., 2015) and chronic lung disease (Beth Israel Deaconess Medical Center, 2014; McGill University, 2016; Oxford Brookes University, 2015). A 2018 double-blind, placebo-controlled trial by Grogono et al. (2018) evaluated inhaled nebulized furosemide (40 mg furosemide in 4 mL 0.9% saline) in healthy adults, demonstrating effective relief of experimentally induced air hunger during dyspnea after multiple dosing per day with no untoward effects. Therefore, accumulated data indicate that furosemide is a cytokine-targeting anti-inflammatory, which may be safely administered by inhalation multiple times per day.

Our in silico screening identified other loop diuretics with structural similarities to furosemide. Bumetanide exhibits anti-inflammatory properties in LPS stimulated RAW264.7 cells, reduces LPS-induced production of cytokines following direct pulmonary administration, and lowers levels of TNF-α production in lung-injured mice (Hung et al., 2018). Bumetanide however failed to inhibit sodium metabisulfite induced bronchoconstriction in asthmatic subjects (O’Connor et al., 1991). Piretanide and azosemide have also been variously studied in models of cytokine-mediated inflammation and bronchoconstriction (Yeo et al., 1992). Since none of these agents are in widespread clinical use, we have elected to pursue the development of furosemide in COVID-19 because of its worldwide availability in the time of a global pandemic.

The potential use of furosemide in the anti-inflammatory treatment of COVID-19 has strengths and weaknesses. Furosemide is inexpensive and available in every country in the world; it is safe and has profound anti-inflammatory cytokine activity, particularly against IL-6 and TNF-α. If administered orally, furosemide can produce a profound diuresis which would be a clinical detriment in a febrile and potentially dehydrated person. When administered orally, the pharmacokinetics of furosemide indicate that it would have primarily intra-vascular distribution, suggesting greater utility early in the course of the disease, but less so in later-stage ventilator-supported individuals in whom macrophage migration from bloodstream to pulmonary tissue has occurred. As the disease progresses, direct administration of furosemide to the lungs by nebulized delivery adequately addresses the need to have furosemide reach intra-alveolar macrophages. Arguably, truly effective treatments may require both anti-viral and anti-inflammatory agents since the early stages of the infection are dominated by high viral replication. However, application of an anti-inflammatory agent only, especially in the current situation where an effective anti-viral is still missing, may have the potential to reduce the number of patients requiring mechanical ventilation and reduce cough as a symptom which could help limiting the spread of the virus.

Care must be taken to prevent viral contamination and bystander exposure during the aerosolized administration of the drug, but this can be achieved with appropriate cautions in place. We are currently pursuing a clinical study of inhaled furosemide in people with COVID-19.

Conclusions

COVID-19 is a pandemic threatening global health. The need to identify innovative therapeutics which may be deployed rapidly and efficiently is a pharmacological priority. Furosemide is a safe, inexpensive, well-studied small molecule which is a reasonably potent inhibitor of IL-6 and TNF-α and may be administered locally to the lungs; pre-clinical data from in silico and in vitro experiments suggest that it may be a candidate for repurposing as an inhaled therapy against the immunopathologies of COVID-19.

Supporting Information

A list of physiochemical descriptors used as initial screening test of the dataset; a list of promising drug candidates along with corresponding physiochemical descriptors; Docking Score ‘S’ of promising drug candidates with IL-6 and TNF-α; Figures and ligand interaction diagrams of mefenamic acid, bumetanide, piretanide and azosemide docking into binding site of TNF-α and IL-6. This information is available free of charge on PeerJ.

Supplemental Information

Supplemental Information 1 Interaction of mefenamic acid and TNF-α.

(A) Binding of mefenamic acid in the active site of TNF-α; (A) Ligand interaction diagram of mefenamic acid in binding site of TNF-α.

Click here for additional data file.

Supplemental Information 2 Interaction of mefenamic acid and IL-6.

(A) Binding of mefenamic acid in the active site of IL-6; (A) Ligand interaction diagram of mefenamic acid in binding site of IL-6.

Click here for additional data file.

Supplemental Information 3 Interaction of bumetanide and TNF-α.

(A) Binding of bumetanide in the active site of TNF-α; (A) Ligand interaction diagram of bumetanide in binding site of TNF-α.

Click here for additional data file.

Supplemental Information 4 Interaction of bumetanide and IL-6.

(A) Binding of bumetanide in the active site of IL-6; (A) Ligand interaction diagram of bumetanide in binding site of IL-6.

Click here for additional data file.

Supplemental Information 5 Interaction of piretanide and TNF-α.

(A) Binding of piretanide in the active site of TNF-α; (A) Ligand interaction diagram of piretanide in binding site of TNF-α.

Click here for additional data file.

Supplemental Information 6 Interaction of piretanide and IL-6.

(A) Binding of piretanide in the active site of IL-6; (A) Ligand interaction diagram of piretanide in binding site of IL-6.

Click here for additional data file.

Supplemental Information 7 Interaction of azosemide and TNF-α.

(A) Binding of azosemide in the active site of TNF-α; (A) Ligand interaction diagram of azosemide in binding site of TNF-α.

Click here for additional data file.

Supplemental Information 8 Interaction of azosemide and IL-6.

(A) Binding of azosemide in the active site of IL-6; (A) Ligand interaction diagram of azosemide in binding site of IL-6.

Click here for additional data file.

Supplemental Information 9 Uncropped Western blot of Fig. 2C, iNOS.png.

Click here for additional data file.

Supplemental Information 10 Uncropped Western blot of Fig. 2C, GAPDH.

Click here for additional data file.

Supplemental Information 11 Uncropped Western blot of Fig. 4A, pro-IL-1β.

Click here for additional data file.

Supplemental Information 12 Uncropped Western blot of Fig. 4A, actin.

Click here for additional data file.

Supplemental Information 13 List of physiochemical descriptors used as initial screening test of the dataset.

Click here for additional data file.

Supplemental Information 14 List of promising drug candidates along with corresponding physiochemical descriptors.

Click here for additional data file.

Supplemental Information 15 Docking Score ‘S’ of promising drug candidates with IL-6 and TNF-α.

Click here for additional data file.

Supplemental Information 16 Raw data of in vitro experiments.

Click here for additional data file.

Additional Information and Declarations

Competing Interests

Author Contributions

Data Availability

The authors declare that they have no competing interests.

Zhiyu Wang conceived and designed the experiments, performed the experiments, analyzed the data, prepared figures and/or tables, authored or reviewed drafts of the paper, and approved the final draft.

Yanfei Wang conceived and designed the experiments, performed the experiments, analyzed the data, authored or reviewed drafts of the paper, and approved the final draft.

Prachi Vilekar conceived and designed the experiments, performed the experiments, analyzed the data, authored or reviewed drafts of the paper, and approved the final draft.

Seung-Pil Yang conceived and designed the experiments, performed the experiments, analyzed the data, authored or reviewed drafts of the paper, and approved the final draft.

Mayuri Gupta conceived and designed the experiments, performed the experiments, analyzed the data, prepared figures and/or tables, authored or reviewed drafts of the paper, and approved the final draft.

Myong In Oh conceived and designed the experiments, performed the experiments, analyzed the data, authored or reviewed drafts of the paper, and approved the final draft.

Autumn Meek conceived and designed the experiments, performed the experiments, analyzed the data, authored or reviewed drafts of the paper, and approved the final draft.

Lisa Doyle conceived and designed the experiments, performed the experiments, analyzed the data, authored or reviewed drafts of the paper, and approved the final draft.

Laura Villar conceived and designed the experiments, performed the experiments, analyzed the data, authored or reviewed drafts of the paper, and approved the final draft.

Anja Brennecke analyzed the data, prepared figures and/or tables, authored or reviewed drafts of the paper, and approved the final draft.

Imindu Liyanage analyzed the data, authored or reviewed drafts of the paper, and approved the final draft.

Mark Reed conceived and designed the experiments, performed the experiments, analyzed the data, authored or reviewed drafts of the paper, and approved the final draft.

Christopher Barden conceived and designed the experiments, performed the experiments, analyzed the data, authored or reviewed drafts of the paper, and approved the final draft.

Donald F Weaver conceived and designed the experiments, performed the experiments, analyzed the data, authored or reviewed drafts of the paper, and approved the final draft.

The following information was supplied regarding data availability:

The raw measurements are available in the Supplemental Files.

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
