# Peer review of "Small molecule therapeutics for COVID-19: repurposing of inhaled furosemide"

_PeerJ, doi:10.7717/peerj.9533_

## Round 0.1 · original submission · Major Revisions

While both reviewers were broadly supportive of the manuscript and the associated findings, they also bring up several issues that need further clarification. In particular, the statistical analysis questions brought up by Reviewer 1 along with those brought up by Reviewer 2 surrounding computational issues should be addressed.

·

Basic reporting

No comment.

Experimental design

No comment

Validity of the findings

No comment

Additional comments

In the present manuscript, Wang and coll. report an interesting in silico / in vitro drug screening strategy for the identification of molecules with multi-target anti-inflammatory effect that could eventually constitute interesting candidates for the treatment of the immune-pathogenic phase of COVID-19. As a result of their screening and in vitro validation approach, they propose furosemide as a possible therapeutic.
The manuscript is very well written, with clear and relevant concepts. The context is well explained and provides the necessary background information for the interpretation of the study and results. The study design is straightforward and the different levels of validation follow a logical sequence. However, I believe that certain points need to be addressed before considering the manuscript for publication in PeerJ.

Major issues:
1) The most important issue to be addressed is related to statistical analysis and the criteria used for data interpretation. On one side, differences observed in Fig 1A-B and 5A-C seem a priori rather minor to support P-values below 0.05 or 0.01, mostly considering error bars. On the other side, the results shown in Fig 4B and do not seem to support categoric statements such as “the amount of secreted IL-1b slightly decreased…” (line 242). I suggest the authors revise the analysis and potentially temperate their conclusions, notably in the case of the putative effect of furosemide on NO production and TNF-a. Including in the supplementary material the dataset from which the analysis was performed would also be desirable.

2) For clarity purposes, I find important to include in Figure 1 a schematic representation of the overall screening strategy prior to the in vitro evaluation, from the +1000 molecules in the library to the identification of the 4 candidate “hits”. This would greatly help the readers to better understand the experimental approach.


Minor issues:
Figures 4, 5, 6: Please consider merging these figures in one to ease analysis/comparison.
Figures 7, 8: Please consider merging these figures in one to ease analysis/comparison. Please provide more information on the methodology in the figure legend.
Figures 9, 10: Please consider merging these figures in one to ease analysis/comparison. Please provide more information on the methodology in the figure legend.
Lines 60-62: I find that the inclusion of the reference “Xu X et al” is rather premature. Although the authors clearly warned on “preliminary data”, this is so far an abstract (the rest of the manuscript is in Mandarin) in a non-peer-reviewed preprint server. I would probably remove this part of the text or ideally find another reference.
Lines 331-333: The authors state that furosemide has “ideal” pk properties for targeting circulating macrophages. Please briefly discuss the potential role of respiratory tissue resident macrophages.
Lines 346-348: “…nebulized furosemide… demonstrating improved efficacy…”. Please explain the context of the trial; i.e. “improved efficacy” against which pathology?
Lines 366-368: The authors propose to use furosemide in the early course of infection. Although I agree that the advanced ICU state would be too late, the early clinical phases of COVID19 are mostly driven by high viral replication, for which the combination with an antiviral seems mandatory.
Discussion (in general): one important limit of the study is that the potential effect of furosemide could not be evaluated in the context of SARS-CoV-2 infection. I do understand the manipulation of the virus is very restricted and of course I do not expect the authors to include this kind of experiments. However, I suggest the authors discuss the potential differences between the LPS-induced inflammatory response model used in this study and those induced by infection.

·

Basic reporting

The main idea of this work was to use both human endogenous substances and a drug dataset, in order to screen for some that could be active against TNF-α and IL-6 cytokines, using computational and in vitro methods. The English language is clear and used the professional terms in all the text. The figures and literature are well-represented in most of them. The authors used suitable literature and background in all sections of the article. The results confirmed the author's hypothesis but, on the other hand, the work needs complementary data in the text results and discussion to be fully considered for publishing. There are other previous articles that used furosemide as an anti-inflammatory agent and combined with other drugs, as can be checked out in my suggestions in the next sections of the review process. On the other hand, the authors bring new insights for furosemide acting in IL-6 and TNF-a. Some improvements are needed in computational methods and results section to become the manuscript acceptable for publication.

Experimental design

The experimental design was well constructed and organized. The methodology was original in using endogenous substances to search for possible compounds with anti-inflammatory action. The authors used the in vitro methods with rigor and ethics, but the computational methods must be better described. See my considerations below:

Lines 78 - 80: Here the authors don’t explain were these endogenous substances and the drug dataset was obtained. Is it from any curated database (PubChem, Drugbank, ZINC)? Please indicate this in the methods section.

Lines 83 - 84: The authors don’t explain how they selected the crystallographic structures for docking. Please include this in the paper.

Lines 92 - 95: Explain why you selected pH 7.4. Do you used it just for the ligands or included the receptors?

Lines: 138 - 139: Please include the concentration of the substrate.

Line 146: Please include polyacrylamide concentration.

Validity of the findings

The work brings new aspects of the immunological action of furosemide, applying to the treatment of COVID-19. In my point of view, these results can be applied to other types of treatments where the patient's inflammatory burden increases suddenly. In this case, a viral infection is just another treatable disease with the help of this medication. The work is innovative in the sense of using computational tools in order to filter endogenous substances with important actions for the treatment of COVID-19, combined with confirmatory in vitro methods. On the other hand, other authors have already used this drug for the same purpose in other diseases. An important point in this work is that the computational results are confirmed by the in vitro tests. My considerations about the findings are described below:

Lines 176 - 178: please indicate from where you obtained these small molecules.

Lines 273 - 277: Please include re-docking for both crystallographic ligands from IL-6 and TNF-a, for validation of your docking findings with 3-HAA.

Lines 287 - 289: What parameters do you have to say that furosemide inhibits TNF-a based on pose energy? I think here you need to include a re-docking with the crystallographic ligand of 2AZ5 (6,7-DIMETHYL-3-[(METHYL{2-[METHYL({1-[3-(TRIFLUOROMETHYL)PHENYL]-1H-INDOL-3-YL}METHYL)AMINO]ETHYL}AMINO)METHYL]-4H-CHROMEN-4-ONE), in order to validate your docking results.

Lines 294 - 296: I think here you need to include a re-docking with the crystallographic ligand of 1ALU (tartaric acid), in order to validate your docking results. Other authors have found anti-inflammatory activity for furosemide, such as Pandota (2002) -  DOI: 10.1097/00045391-200207000-00009 ; Berti et al. (1995) - https://pubmed.ncbi.nlm.nih.gov/8595068/. Then, in this line, the authors do not provide significant new findings without completely validate the biochemical mechanism of action of furosemide in this case. Please, include re-docking for validating your computational findings.

Additional comments

It is an original and well-organized work but lacks some computational methods information and validations which will improve the quality of the work.

---

## Round 0.2 · accepted · Accept

Thank you for addressing reviewer questions and congratulations again.

·

Basic reporting

No comment

Experimental design

No comment

Validity of the findings

No comment

Additional comments

The authors satisfactorily addressed my major concerns. I would like to thank them for the effort. I therefore consider this new version of the manuscript acceptable for publication.

·

Basic reporting

No comment

Experimental design

No comment

Validity of the findings

No comment

Additional comments

Thank you for all changes in the paper and for following the suggestions.